# Prokaryotic Expression of *Eimeria magna* SAG10 and SAG11 Genes and the Preliminary Evaluation of the Effect of the Recombinant Protein on Immune Protection in Rabbits

**DOI:** 10.3390/ijms231810942

**Published:** 2022-09-19

**Authors:** Jiayan Pu, Jie Xiao, Xin Bai, Hao Chen, Ruoyu Zheng, Xiaobin Gu, Yue Xie, Ran He, Jing Xu, Bo Jing, Xuerong Peng, Guangyou Yang

**Affiliations:** 1Department of Parasitology, College of Veterinary Medicine, Sichuan Agricultural University, Wenjiang, Chengdu 611130, China; 2Department of Chemistry, College of Life and Basic Science, Sichuan Agricultural University, Wenjiang, Chengdu 611130, China

**Keywords:** rabbit, *Eimeria magna*, SAG10, SAG11, recombinant protein, immune protection

## Abstract

*Eimeria magna* is a common coccidia in the intestines of rabbits, causing anorexia, weight loss, diarrhea, and bloody stools. This study cloned and determined the expression levels of four Eimeria surface antigens (EmSAGs) at different developmental stages and showed that *EmSAG10* and *EmSAG11* are highly expressed at the merozoite stage. Rabbits were immunized with r*EmSAG10* and r*EmSAG11*, and then challenged with *E. magna* after 2 weeks. Serum-specific antibodies and cytokine levels were detected using ELISA. Immune protection was evaluated based on the rate of the oocysts decrease, the output of oocysts (*p* < 0.05), the average weight gain, and the feed: meat ratio. Our results showed that rabbits immunized with r*EmSAG10* and r*EmSAG11* had a higher average weight gain (62.7%, 61.1%), feed; meat ratio (3.8:1, 4.5:1), and the oocysts decrease rate (70.8%, 81.2%) than those in the control group, and also significantly reduced intestinal lesions. The specific IgG level increased one week after the first r*EmSAG10* and r*EmSAG11* immunization and was maintained until two weeks after the challenge (*p* < 0.05). The TGF-β, IL-4, and IL-10 levels in the serum increased significantly after the secondary immunization with r*EmSAG10* and r*EmSAG11*, while the IL-2 levels increased significantly after the secondary immunization with r*EmSAG11* (both *p* < 0.05), suggesting that r*EmSAG10* can induce a humoral and cellular immunity, while r*EmSAG11* can only induce a humoral immunity. Therefore, r*EmSAG10* is a candidate antigen for *E. magna* recombinant subunit vaccines.

## 1. Introduction

*Eimeria**magna* is a common coccidia in rabbit intestines. It mainly parasitizes the jejunum and ileum of rabbits. It causes anorexia, weight loss, and diarrhea in rabbits. Infection leads to bloody stools and seriously affects rabbit reproduction [1,2,3,4,5,6]. Currently, the control of rabbit coccidiosis relies mainly on anticoccidial drugs. Although it is inexpensive to add anticoccidial drugs to feed, drug resistance and drug residues remain major problems. Live, attenuated vaccines have good anticoccidial effects, but in some cases, there is a risk of restoring virulence [7,8]. The recombinant subunit vaccine is safer, stable, easy to make in large quantities, and costs less than the live attenuated vaccine [9].

Coccidia have a complex life history, and their antigen composition and distribution are inconsistent across different developmental stages [10]. The immunogenicity of coccidia in the early stages of endogenesis is stronger than in the later stages of sexual reproduction [10]. Located on the surface of coccidia during invasion, the Eimeria surface antigens (SAGs) family is large, and their structural characteristics are obvious [11,12,13]. Recombinant chicken coccidia SAGs can effectively reduce intestinal lesions and weight loss, which have certain immune-protective effects [14,15,16]. Some attenuated strains of *E. magna* in rabbits have been studied [17,18,19,20,21], but there is no information on recombinant subunit vaccines for this species. In the present study, we screened four SAGs from the transcriptome of *E. magna*. By analyzing the transcription levels of the four genes at different developmental stages of *E. magna*, SAG10, and SAG11 were selected and expressed. The immunoprotective effects of the recombinant proteins were evaluated in animal experiments. This study aimed to provide a reference for the development of recombinant subunit vaccines for *E. magna*.

## 2. Results

### 2.1. Cloning and Bioinformatics Analysis of EmSAGs Genes

SAG1 (GenBank accession number: ON468435), SAG2 (GenBank accession number: ON468436), SAG10 (GenBank accession number: OM897224), and SAG11 (GenBank accession number: OM897225) were amplified from *E. magna* cDNA. The amplified fragments were 606 bp, 126 bp, 648 bp, and 678 bp, respectively. Following the cloning and sequencing analysis, the sequence similarity of each amplified fragment was 100% with that of EmSAG1, EmSAG2, EmSAG10, and EmSAG11 in the transcriptome data of *E. magna.*

### 2.2. Bioinformatics Analysis

There is no signal peptide region and transmembrane helix region in *EmSAG1*, and the GPI anchor site is amino acid 174 (Table 1). A multiple sequence comparison of *EmSAG1* and other *Eimeria* coccidia showed that the similarity ranged from 22.22%–33.83% (Figure 1).

There was no signal peptide region, transmembrane helix region and GPI anchor site in *EmSAG2* (Table 1). The multiple sequence alignment is invalid because its amino acid sequence is too short.

There was no signal peptide region in *EmSAG10*, and its transmembrane region was a 195–214 amino acid sequence. The GPI anchor site is amino acid 193 (Table 1). *EmSAG10* encoding a protein with a predicted MW of 41 kDa. The multiple sequence comparison of *EmSAG10* and other *Eimeria* coccidia showed that the similarities ranged from 23.4%–34.3% (Figure 1).

The amino acid sequence of the *EmSAG11* signal peptide was a 1–23 amino acid sequence, and its transmembrane domain was a 224–246 amino acid sequence. The GPI anchor site is amino acid 225 (Table 1). *EmSAG11* encoding a protein with a predicted MW of 42 kDa. The multiple sequence comparison of *EmSAG11* and other *Eimeria* coccidia showed that the similarity ranged from 26.47%–36.73% (Figure 1).

### 2.3. Analysis of the Transcription Level Differences in Different Development Stages of EmSAGs

The expression of *EmSAG1 (p* < 0.001) and *EmSAG2 (p* < 0.0001) in the sporidized stage of *E. magna* was higher than that in other stages (Figure 2), and the expression of *EmSAG10 (p* < 0.0001) and *EmSAG11 (p* < 0.0001) in the merozoite stage was higher than that in the other stages (Figure 2), the difference was extremely significant (*p* < 0.0001).

### 2.4. Expression of rEmSAG10 and rEmSAG11

The successfully constructed prokaryotic expression recombinant plasmids of the two genes were transformed into *E. coli* BL21 (DE3) competent cells for the induction of the expression. pET-32a is a tagged fusion protein and we saw that a band of about 40 kDa for r*EmSAG10* and r*EmSAG11*, which contained the signature fusion protein was about 20 kDa, and the predicted sizes were 21 kDa and 22 kDa, respectively, so the predicted recombinant proteins were 41 kDa and 42 kDa, and our results were consistent with the predicted size. Both pET-32a (+) -r*EmSAG10* and pET-32a (+) -r*EmSAG11* were expressed in the inclusion bodies. pET-32a (+) -r*EmSAG10* (Figure 3a) was best expressed at 37 °C, and pET-32a (+) -r*EmSAG11* (Figure 3b) was induced at different temperatures and was best expressed at 18 °C. The induction results at different time periods showed that the induction effect improved with an increase in time in the range of 10 h.

### 2.5. Western Blotting of rEmSAG10 and rEmSAG11

A western blot analysis showed that the rabbit serum infected with *E. magna* could recognize r*E*m*SAG10* and r*EmSAG11* (Figure 4, lanes 1 and 2), and the pET-32a tag protein could not be recognized by the rabbit serum infected with *E. magna* (Figure 4, lane 3), whereas the rabbit-negative serum did not recognize r*EmSAG10* or r*EmSAG11* (Figure 4, lanes 4 and 5).

### 2.6. Protective Efficacy of rEmSAG10 and rEmSAG11

Following the first and second immunizations, there were no adverse reactions in the experimental rabbits in each group. According to the average weight gain after the immunization before the challenge, there was no significant difference between the groups (*p* > 0.05), indicating that r*EmSAG10* and r*EmSAG11* were in our experiment. It has a good safety profile at the doses used (Table 2).

None of the rabbits in the control group died after infection, with symptoms of mental depression, anorexia or obvious diarrhea. The typical pathological changes were observed in the middle and lower segments of the small intestine. The intestine was filled with mucus, blood vessels in the intestinal wall, congestion, and bleeding in the intestinal mucosa (Figure 5a–c); whereas in the unchallenged control group, there were no lesions in the middle and lower segments of the small intestine (Figure 5d). In the immune groups of r*EmSAG10* and r*EmSAG11*, no rabbit died after infection, the rabbits were active, had a normal appetite, and did not have diarrhea symptoms. An autopsy showed a small number of hemorrhagic filaments in the r*EmSAG10* and r*EmSAG11* groups (Figure 5e), and there were a few bleeding spots in the crypt of the intestinal tract in the immune group of r*EmSAG11* (Figure 5f).

Compared with the control group, the relative weight gain rates of r*EmSAG10* and r*EmSAG11* immunized rabbits were 62.7% and 61.1%, respectively (*p* > 0.05) (Table 2). In addition, the oocyst yield in rabbits immunized with r*EmSAG10* (70.8% oocyst decrease rate) and r*EmSAG11* (81.2% oocyst decrease rate) was significantly decreased (*p* < 0.05). Compared with the challenged control (6.1:1), quil-A control (5.9:1) and recombinant pET-32a tag protein control (6:1) positive control group, r*EmSAG10* (3.8:1) and r*EmSAG11* (4.5:1) were the immunized group had a better feed conversion ratio.

### 2.7. Detection of Specific Antibody Levels

Compared with the control group (challenged control, Quil-A saponin, Recombinant pET-32a tag protein), the average specific IgG levels in the serum of the r*EmSAG10* (Figure 6a) and r*EmSAG11* (both *p* < 0.05) (Figure 6b) immunized groups reached a higher level one week after the first immunization, and the specific IgG level after the second immunization tended to be stable, which could be maintained at a higher level until two weeks after the challenge.

### 2.8. Cytokine Detection

Following the second immunization, there were significant differences in IL-2, IL-4, IL-10, and TGF-β concentrations between r*EmSAG10* and the control group (PBS-infected, Quil-A saponin, Recombinant pET-32a tag protein) (*p* < 0.05). There was no difference in the IL-17 and IFN-γ concentrations (*p* > 0.05) (Figure 7).

There were significant differences in IL-4, IL-10, and TGF-β concentrations between r*EmSAG11* and the control group (PBS-infected, Quil-A saponin, Recombinant pET-32a tag protein) (*p* < 0.05). There were no differences in IL-2, IL-17, or IFN-γ concentrations (*p* > 0.05) (Figure 7).

## 3. Discussion

In this study, SAG10 and SAG11, which are highly expressed in the merozoite stage of *E. magna*, were selected, and then the r*EmSAG10* and r*EmSAG11* proteins were obtained as subunit vaccines through the prokaryotic expression. The animal experiments showed that r*EmSAG10* and r*EmSAG11* immunization led to average weight gain and decreased oocyst output compared to the control group (PBS-infected, Quil-A saponin, Recombinant pET-32a tag protein). The relative weight gain rates of r*EmSAG10* and r*EmSAG11* were 62.7% and 61.1%, respectively, and their oocyst decrease rate were 70.8% and 81.2%, respectively. The results showed that they had a protective effect.

*Eimeria magna* is an intestinal coccidia found in rabbits. It mainly invades intestinal epithelial cells, which causes intestinal bleeding and huge economic losses to the rabbit industry. Compared with drug prevention and live vaccines, subunit vaccines are a promising alternative strategy for control [22,23,24]. In recent years, subunit vaccines of chicken coccidia, such as MIC2, GAM56, and SAG4, have shown better protective effects [25,26,27].

A surface antigen is one of the candidate antigens for the development of a new generation of vaccines [15,16,28]. Its core function may be to attach itself to host cells before the parasite invasion [29], and the early infection of parasites on host cells can stimulate the host to produce a good immune response [30,31]. Researchers inoculated chickens with r*EtSAG* to induce humoral and cellular immunity and to protect against *E. tenella* [32]. Subsequent studies have also confirmed that r*EtSAG*, as a recombinant vaccine, can induce high levels of antibodies in the host and has a certain effect on chicken coccidiosis infection [33]. The invasion of *E. tenella* SAG mainly manifests in the sporozoite and merozoite stages [34].

IFN-γ and IL-2 are Th1-type cytokines and play an important role in the resistance to *Eimeria* infection [35]. IL-2 promotes the growth and differentiation of many immune cells [36,37]. In the present study, compared with the control group, the secretion level of IL-2 in the experimental rabbits inoculated with r*EmSAG10* was significantly increased (*p* < 0.05), indicating that the anticoccidial immunity was activated in the experimental rabbits inoculated with r*EmSAG10*. However, the secretion level of IFN-γ is not significant. However, the secretion level of IFN-γ did not change significantly in this study. The reason for this phenomenon may be that the increase of IL-10 (Th2 cytokine) promotes the occurrence of the Th2 response, thereby inhibiting the secretion of IFN-γ [38,39,40].

The Th2-type cytokines, mainly IL-4, can regulate humoral immunity, activate helper B cells [41,42], and as other studies have shown, elevated IL-4 levels after inoculation with various recombinant proteins [43,44]. In this study, compared with the control group, we detected IL-4 levels in the serum after two immunizations with r*EmSAG10* and r*EmSAG11*, and observed that IL-4 in the serum was significantly increased, and the antibody response in this study showed that after inoculation with r*EmSAG10* and r*EmSAG11*, the level of IgG in the serum was significantly increased, and both IL-4 and IgG increased, indicating that r*EmSAG10* and r*EmSAG11* can induce humoral immunity, thereby playing a certain role in the resistance to *Eimeria* infection.

TGF-β is produced by regulatory T cells and it is an important inflammatory regulator that exhibits pro-inflammatory properties at low concentrations and anti-inflammatory effects at high concentrations [45,46], and stimulates the repair of the damaged mucosal epithelial integrity following injury [47]. IL-10 can inhibit host injury and reduce the production of pro-inflammatory cytokines [48,49,50,51]. In this study, the expression of IL-10 and TGF-β in r*EmSAG10* and r*EmSAG11* after the second immunization was compared, indicating that r*EmSAG10* and r*EmSAG11* can inhibit the occurrence of inflammation, reduce host intestinal damage, and thus suggests that TGF-β and IL-10 play a role not only in *E. magna* infection, but also in the immune response to *E. magna* antigens.

In summary, *EmSAG10*, and *EmSAG11* are highly expressed during the merozoite stage. The specific IgG antibody levels in the serum of rabbits immunized with r*EmSAG10* and r*EmSAG11* were still high two weeks after the challenge, and TGF-β, IL-2, IL-4, and IL-10 levels were significantly increased in the serum after the second immunization with r*EmSAG10*, and TGF-β, IL-4, and IL-10 levels were significantly increased in the serum after the second immunization with r*EmSAG11*, suggesting that rEmSAG10 can induce humoral and cellular immunity. The results of animal experiments also showed that r*EmSAG10* could increase the average body weight and decrease the output of oocysts, indicating that r*EmSAG10* is more suitable as a candidate antigen for recombinant subunit vaccines of *E. magna*.

## 4. Materials and Methods

### 4.1. Animals, Parasites, and Serum

The Beijing strain of *E. magna* was provided by Professor Xianyong Liu of China Agricultural University. It was subcultured and preserved at the Department of Parasitology of Sichuan Agricultural University. Four stages of *E. magna*, including unsporulated oocysts, sporulated oocysts, merozoites, and gametophytes, were isolated from a rabbit artificially infected with *E. magna.*

Forty-eight 35-day-old coccidian-free New Zealand rabbits (850 ± 97.7 g, 24 males and 24 females) were provided by the Department of Parasitology, College of Veterinary Medicine, Sichuan Agricultural University and kept strictly in a coccidian-free environment according to the method described by Shi et al. [52]. The young rabbits were weaned at the age of 18 days, and fed high-temperature sterilized feeding pellets (in-house prepared at this laboratory) in combination with human infant formula until 30 days of age, during which boiled drinking water and feed were provided *ad libitum*. Anticoccidial drugs were used in cross-rotation in water, and rabbit cages were regularly flame sterilized to prevent contamination with other *Eimeria* species.

Negative and positive serum samples of *E. magna* were provided by our laboratory.

### 4.2. Total RNA Extraction and Reverse Transcription

The total RNAs from the four stages of *E. magna* were extracted using the RNA Extraction Kit (TaKaRa, Dalian, China) and reverse transcribed into cDNA using a reverse transcription system kit (PrimeScript™ RT reagent kit with gDNA Eraser, TaKaRa, Dalian, China), and stored at −80 °C until use.

### 4.3. Cloning and Bioinformatics Analysis of the EmSAGs Genes

The sequences of *EmSAG1*, *EmSAG2*, *EmSAG10*, and *EmSAG11* were obtained by cloning and sequencing, and a bioinformatic analysis was performed using ExPASY (http://web.expasy.ory/protparam/ accessed on 28 May 2022).The physicochemical properties of the proteins were predicted, including the number of amino acids, molecular weight, and isoelectric point; the protein signal peptide, transmembrane region, secondary structure, GPI anchor site, and B antigen linear epitope using SignalP-5.0 (https://www.novopro.cn/tools/signalp.html, accessed on 28 April 2022), TMHMM-2.0 (http://www.cbs.dtu.dk/services/TMHMM-2.0, accessed on 28 April 2022), Jaview (https://webs.iiitd.edu.in/raghava/apssp2/, accessed on 28 April 2022), (http://mendel.imp.ac.at/gpi/plant-server, accessed on 28 April 2022), (http://tools.immuneepitope.org/bcell/ accessed on 28 April 2022) Forecast. In addition, multiple sequence alignments of *EmSAG1*, *EmSAG2*, *EmSAG10*, and *EmSAG11* and their homologous genes were performed using the DNAMAN software.

### 4.4. Transcription Level of the SAGs Genes

The cDNA was prepared by the above-mentioned method to obtain the unsporidized, sporidized, merozoites, and gametophytes stages of rabbit *E. magna* as a template, and *EmSAG1*, *EmSAG2*, *EmSAG10*, and *EmSAG11* (Table 3) genes were amplified by real-time quantitative PCR (Roche, Switzerland), and *β-Tubulin* and *GAPDH* (Table 3) were the two housekeeping reference genes for comparison. A 20-µL reaction system consisted of 10 μL TB Green *Premix Ex Taq*II (Tli RNaseH Plus) (2X) (TaKaRa, Dalian, China), 0.8 μL Forward primer (Sangong, Shanghai, China), 0.8 μL Reverse primer (Sangong, Shanghai, China), 2 μL cDNA template, and 6.4 μL ddH2O. The procedure was as follows: one-step 95 °C for 30 s; two-step 95 °C for 5 s, 60 °C for 30 s; 40 cycles, 95 °C for 15 s, 60 °C for 60 s, and 95 °C for 15 s. Three parallel repeats were performed for each of the four genes. We screened two stable reference genes, *Em*-GAPDH and *Em*-β-tubulin, and *Em*-β-tubulin was selected as the reference gene due to its stability among the different developmental stages; we took the unsporulated stage as the calibrator, and by subtracting the expression level of *Em*-β-tubulin, we were able to compare the expression level of other developmental stages. The relative transcription levels of the four genes at each developmental stage of *E. magna* were analyzed using the △△Ct (Qr = 2^-^^△△Ct^) method [53], and subsequently analyzed by one-way ANOVA with a Tukey’s post-hoc test using GraphPad Prism version 8.4.

### 4.5. Prokaryotic Expression and Purification

The cDNA of *E. magna* prepared using the above method was used as a template, and specific primers were designed with reference to the full-length coding sequences of the *EmSAG10* and *EmSAG11* genes (Table 4).

The extracted *E. manga* cDNA was used as a template for the amplification. The reaction conditions were as follows: 95 °C for 3 min; 35 cycles of 95 °C for 45 s, 60 °C for 45 s, 72 °C for 30 s; 72 °C for 10 min; and 4 °C for 5 min. The PCR products were ligated into the pMD-19 T vector (TaKaRa, Dalian, China) and transformed into DH5α competent cells (Tiangen, Beijing, China). Once a single colony was identified as positive by PCR, the plasmids were extracted and sent to Shanghai Bioengineering Co., Ltd. for sequencing. Once sequencing was correct, the above four were subjected to a fluorescence quantitative PCR to select *EmSAG10* and *EmSAG11*, which were highly expressed in the large merozoite stage, and then the successful construction of pMD19-T-*Em*-SAG10 and pMD19-T-*Em*-SAG11 plasmid was subcloned into pET-32a(+) vector (Tiangen, Beijing, China) after the double-enzyme digestion. The constructed pet-32a (+) -*Em*-SAG10 and pet-32a (+) -*Em*-SAG11 were transformed into *E. coli* BL21 (DE3) (TianGen, Beijing, China) and induced with IPTG (Sigma, Cream Ridge, NJ, USA) at 37 °C for 12 h. The cells were collected and sonicated, and SDS-PAGE analyzed the supernatant and precipitate to verify the expression of the recombinant protein. The recombinant proteins were purified using an Ni2-affinity chromatography column (Qiagen, Dusseldorf, Germany), and the purity was detected by 12% SDS-PAGE (Solebao, Beijing, China).

### 4.6. Western Blot Analysis of rEmSAG10 and rEmSAG11

The purified rEmSAG10 and rEmSAG11 were subjected to SDS-PAGE and transferred to a nitrocellulose membrane (Kemaijie, Chengdu, China) using a semi-dry membrane transfer tank (Bio-Rad Laboratories, Hercules, CA, USA). Once cleaning the membrane in TBST three times (5 min each time), the membrane was sealed in TBST with 5% skimmed milk powder (Sangong, Shanghai, China) at 4 °C for 2 h. Once the sealing was completed, the positive and negative serums of *E. magna* (dilution 1:100) were added, and incubated at 4 °C overnight. Following the washing of the membrane with TBST three times, it was incubated with horseradish peroxidase-labeled goat anti-rabbit IgG (HRP-IgG) (Boster, Wuhan, China) at a dilution of 1:1000 at room temperature for 2 h. Following the washing of the membrane with TBST five times, the reaction was stopped with a diaminobenzidine (DAB) substrate solution (Tiangen, Beijing, China).

### 4.7. Immunization Procedure and Experimental Grouping

A subcutaneous injection into the neck was used for immunization. The immunization and vaccination statuses of each group are shown in Table 5.

The rabbits were immunized for the first time at 42 days of age and then immunized with the same dose 14 days later. The blood samples were collected and stored at −20 °C, The body weight was recorded every 7 days. Except for the unchallenged control group, each rabbit in each group was orally inoculated with 1 × 10^5^ newly collected sporulated oocysts. Each experimental rabbit was infected with 100 g of anticoccidial-free feed every day.

### 4.8. Evaluation of the Protective Efficacy of rEmSAG10 and rEmSAG11

Safety observation: The body weight was measured at the time of the first immunization, second immunization, and the challenge, and the health status of all experimental rabbits was observed after immunization.

Following infection, the mental state, appetite for food, and diarrhea were observed, and all experimental rabbits were sacrificed 14 days after infection. The lesions of the jejunum and ileum were observed, photographed and recorded, and the relative weight gain rate, feed to meat ratio, and the oocyst decrease rate were measured. According to the national standard for the diagnosis of coccidiosis in animals (GB/T18647-2020), the appropriate amount of rectal fecal samples were collected for the quantitative examination of oocyst output. The average oocysts per gram (OPG) of each group was counted, and the reduction rate of the oocysts was calculated.

The average weight gain before the challenge was equal to the body weight before the first immunization. The average weight gain after the challenge was equal to the body weight before the sacrifice and before the challenge. The relative weight gain rate = (average weight gain of each experimental group/average weight gain of unchallenged group) × 100%; The rate of feed to meat = (total feed before feeding − total amount of residual feed after sacrificed)/total weight gain after infection; Output of oocysts = experimental group OPG/number of animal; Oocyst decrease rate (%) = (challenged control group OPG -immune group OPG)/(challenged control group OPG) × 100%.

### 4.9. Detection of the Serum Antibody IgG

The serum was collected by weekly blood sampling before the first vaccination until the sacrifice period, and the levels of the specific IgG of r*EmSAG10* and r*EmSAG11* were determined by indirect ELISA, and the serum samples (1:100 dilution) of goat anti-rabbit IgG (Doctor De, Wuhan, China) were detected by the HRP labeled secondary antibody.

### 4.10. Cytokine Detection

An ELISA kit (CUSABIO, Wuhan, China) was used to determine the concentrations of IL-2, IL-4, IL-10, IL-17, TGF-β, and IFN-γ in the serum of the experimental rabbits after the second immunization (before the challenge) according to the instructions.

### 4.11. Data Analysis

All data were analyzed using SPSS software (IBM SPSS version 20.0; SPSS Inc., Chicago, IL, USA) using one-way analysis of variance and the Duncan post-hoc multiple range test. There was no significant difference between the different experimental groups when *p* > 0.05, but there was a significant difference when *p* < 0.05. The difference was extremely significant (*p* < 0.01). All data are expressed as mean ± standard deviation (SD). Graphs were generated using GraphPad Prism 8.4 software.

## 5. Conclusions

Our results indicate that r*EmSAG10* can induce a humoral and cellular immunity and have certain immune protective effects, and r*EmSAG10* can be used as a candidate antigen for recombinant subunit vaccines of *Eimeria magna* in rabbits.

## Figures and Tables

**Figure 1 ijms-23-10942-f001:**
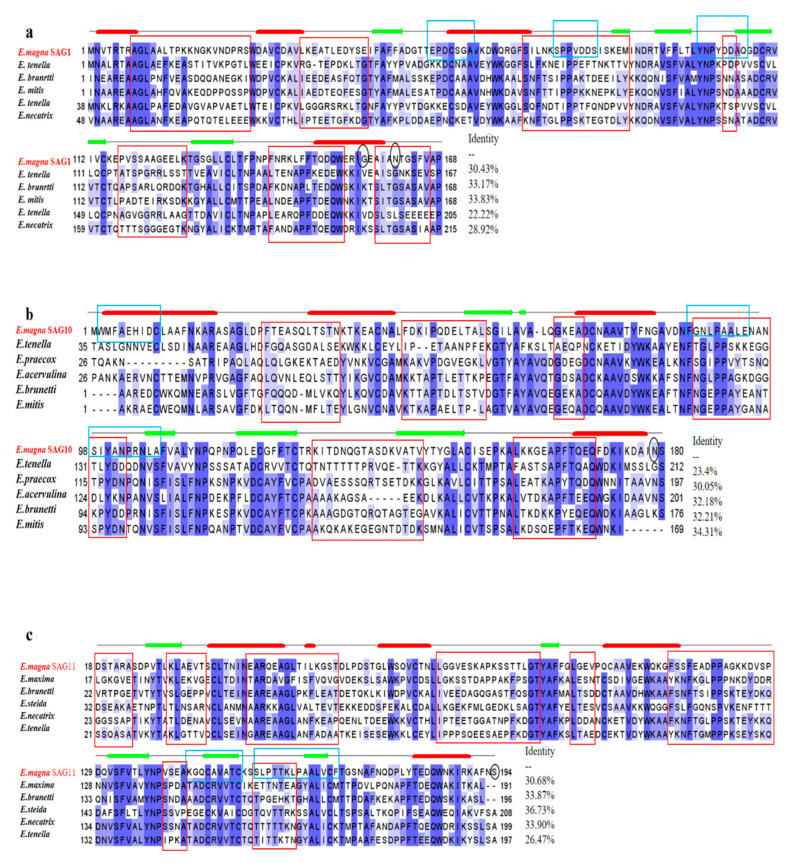
Multiple sequence alignments of SAG1 (**a**), SAG10 (**b**), and SAG11 (**c**) from different *Eimeria*. (**a**) Multiple alignment of *E. magna* SAG1 with SAG proteins from its *Eimeria*: *Eimeria tenella* (UniProt: XP_013232730), *Eimeria brunetti* (UniProt: CDJ45679.1), *Eimeria mitis* (UniProt: XP_037878758.1), *Eimeria tenella* (UniProt: CAE52292.2), *Eimeria necatrix* (UniProt: XP_013439715.1); (**b**) Multiple alignment of *E. magna* SAG10 with its SAG protein from *Eimeria*: *Eimeria tenella* (UniProt: CAE52313.1), *Eimeria praecox* (UniProt: CDI74849.1), *Eimeria acervulina* (UniProt: XP_013251483.1), *Eimeria brunetti* (UniProt: CDJ51402.1), *Eimeria mitis* (UniProt: XP_013355552.1); (**c**) *E. magna* Multiple alignment of SAG11 with SAG proteins from its *Eimeria*: *Eimeria maxima* (UniProt: XP_013334600.1), *Eimeria brunetti* (UniProt: CDJ45678.1), *Eimeria stiedai* (GenBank accession number: QIS60152.1), *Eimeria necatrix* (UniProt: XP_013439715.1), *Eimeria tenella* (UniProt: CAE52299.1). Blue shading indicates the conserved residues. Red boxes represent B-cell epitopes. Blue boxes indicate T-cell epitopes of *E. magna* SAGs. Black circles indicate GPI anchor points.

**Figure 2 ijms-23-10942-f002:**
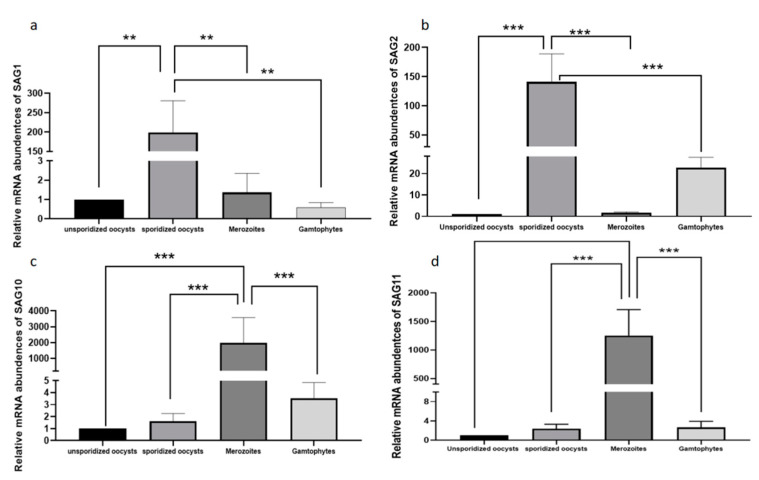
The relative transcriptional levels of EmSAGs at different developmental stages. (**a**) is the relative transcriptional levels of *EmSAG1* at different developmental stages; (**b**) is the relative transcriptional levels of *EmSAG2* at different developmental stages; (**c**) is the relative transcriptional levels of *EmSAG10* at different developmental stages; (**d**) is the relative transcriptional levels of *EmSAG11* at different developmental stages; Data are shown as mean  ±  SD of three replicates per group. ** *p*  <  0.01, *** *p*  <  0.001. All values were estimated by ANOVA (Duncan’s post hoc) at *p* ≤ 0.05.

**Figure 3 ijms-23-10942-f003:**
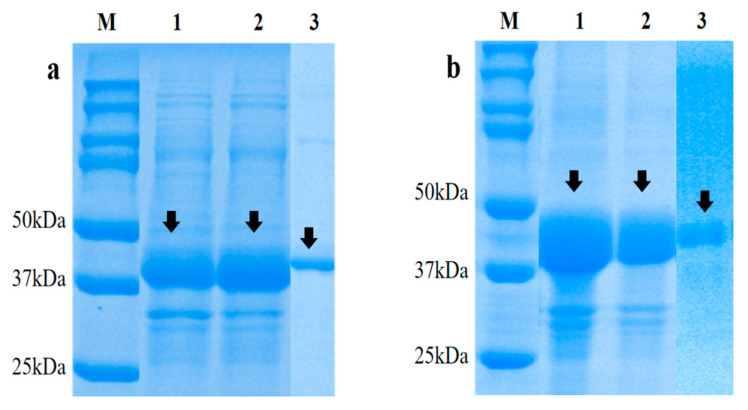
Expression and purification of r*Em*SAG10 (**a**) and r*Em*SAG11 (**b**). M: Bio-red protein molecular weight standard; lanes 1 and 2 in (**a**) are r*Em*SAG10 inclusion bodies for the solubility analysis; lane 3 in (**a**) is purified r*Em*SAG10. Lanes 1 and 2 in (**b**) are r*Em*SAG11 inclusion bodies for the solubility analysis, and lane 3 in (**b**) is purified r*Em*SAG11.

**Figure 4 ijms-23-10942-f004:**
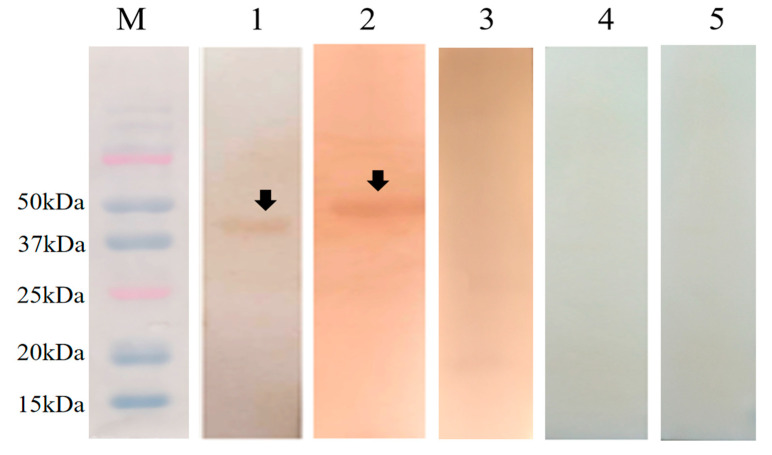
Western blotting analysis of r*EmSAG10* and r*EmSAG11.* M: Bio-red protein molecular weight standard; lanes 1 and 2 were the identified as r*EmSAG10* and r*EmSAG11* in the rabbit serum infected with *E. magna*; Lane 3 was the identified as the pET-32a tag protein protein in the rabbit serum infected with *E. magna*; lanes 4 and 5 were identified as r*EmSAG10* and r*EmSAG11* in the rabbit-negative serum for *E. magna* without infection.

**Figure 5 ijms-23-10942-f005:**
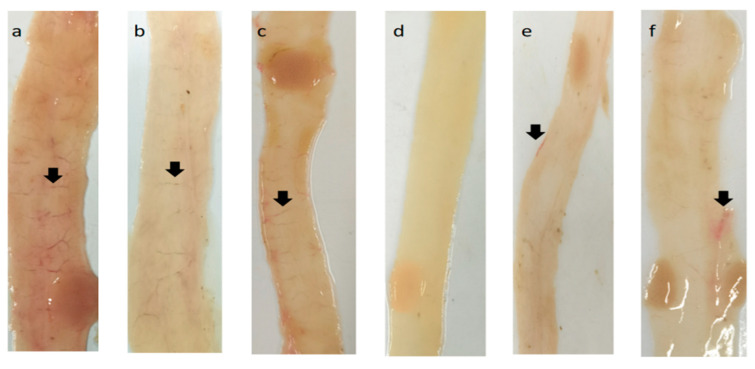
Dissection of the middle and lower segments of the small intestine in each group. (**a**), challenged control group; (**b**), recombinant pET-32a tag protein control group; (**c**), quil-A saponin control group; (**d**), intestinal tract of the blank control group; (**e**), r*Em*SAG10 immune group; (**f**), r*Em*SAG11 immunized group. Note: The black arrow indicates intestinal bleeding or congestion.

**Figure 6 ijms-23-10942-f006:**
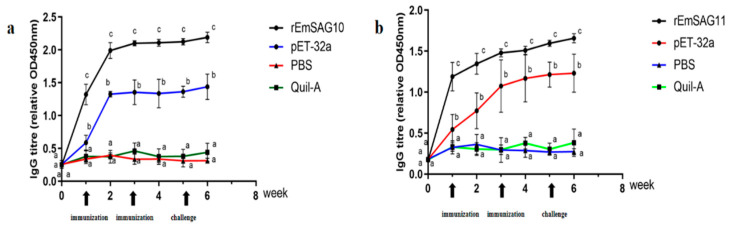
Changes of specific antibody levels in the serum of r*EmSAG10* (**a**) and r*EmSAG11* (**b**). Immunization: first and second immunizations with the vaccine, Challenge: challenge with *E. mange*. (**a**), Detecting the IgG antibody in the vaccination groups and the control group by r*EmSAG10* proteins-based ELISA. (**b**), Detecting the IgG antibody in the vaccination groups and control group by r*EmSAG11* proteins-based ELISA. All groups were challenged with 1 × 10^5^ sporulated oocysts of *E. magna*. Different superscripts in small letters (a, b, c) indicate significant differences in the serum IgG values within each group over time. All values were estimated by ANOVA (Duncan’s post hoc) at *p* ≤ 0.05.

**Figure 7 ijms-23-10942-f007:**
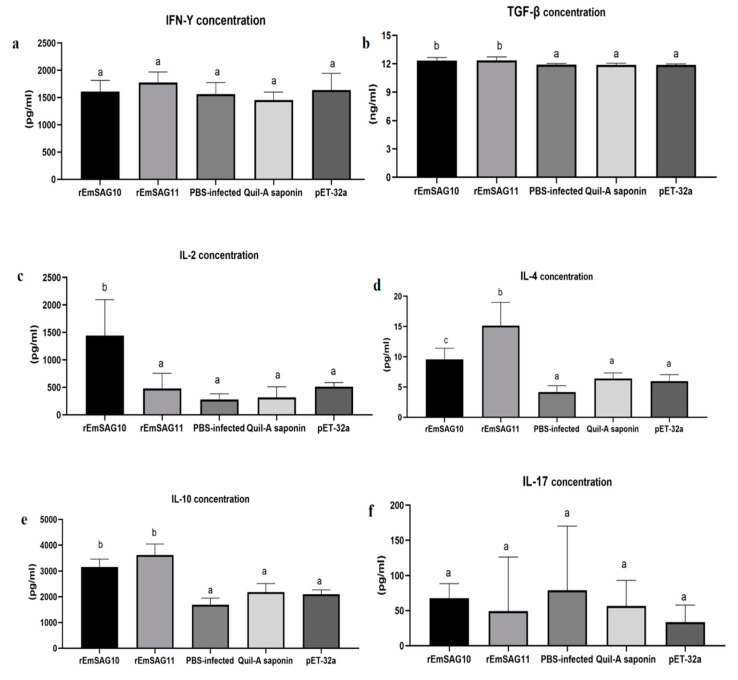
Measurement of the cytokines in the serum of r*Em*SAG10 and r*EmSAG11* before infection. The levels of cytokines IFN-γ (**a**), TGF-β (**b**), IL-2 (**c**), IL-4 (**d**), IL-10 (**e**), and IL-17 (**f**) in the rabbit serum were measured using ELISA. Different superscripts (a, b) indicate a significant difference (*p* < 0.05). The same superscript indicates no significant difference (*p* > 0.05). The unit of TGF-β concentration was ng/mL, and the unit of other cytokines concentration was pg/mL. All values were estimated by ANOVA (Duncan’s post hoc) at *p* ≤ 0.05.

**Table 1 ijms-23-10942-t001:** Analysis of the basic molecular characteristics of SAGs.

Genes	Number of Amino Acids	Molecular Weight	Isoelectric Point	Signal Peptide	Transmembrane Area	GPI Anchor Point	Linear Epitope of B Antigen
SAG1	201	21.76	4.48	-	-	174	5–33, 38–55, 63–72, 79–102, 114–118, 127–146, 157–184
SAG2	41	4.58	10.66	-	-	-	6–14, 16–18, 21, 23–38
SAG10	215	22.81	4.84	-	195–214	193	6–9, 18–37, 45–58, 68–72, 85–106, 119–123, 133–156, 168–186, 189–197
SAG11	247	26.07	4.94	1–23	224–246	225	17–34, 43–77, 86–101, 108–126, 129–156, 167–171, 180–190, 202–217, 220–228

**Table 2 ijms-23-10942-t002:** Protective effects of r*EmSAG10* and r*EmSAG11* against an *E. magna* infection under different evaluation indicators.

Group	Weight Gain after Immunization and before Challenge	Gain Weight after Challenge	Relative Weight Gain Rate (%)	Feed Meat Ratio	Output of Oocysts	Oocyst Decrease Ratio
Unchallenged control	579.3 ± 117.7 ^a^	503.1 ± 126 ^b^	100	2.7:1	0	0
Challenged Control	626.8 ± 133.4 ^a^	226.8 ± 147.8 ^a^	50.3	6.1:1	18,491.6 ± 14,467.0 ^a^	0
Quil-A Saponin	678.7 ± 200.1 ^a^	236.8 ± 81.5 ^a^	45.0	5.9:1	18,216 ± 106.8 ^a^	13.1%
recombinant pET-32a tag protein	629.3 ± 156.6 ^a^	232.5 ± 104.1 ^a^	46.2	6:1	16,483.3 ± 8410.4 ^a^	−11.6%
r*EmSAG10* immunized	602.5 ± 62.7 ^a^	325.4 ± 78.8 ^b^	62.7	3.8:1	5400 ± 78.9 ^b^	70.8%
r*EmSAG11* immunized	646.2 ± 103.5 ^a^	307.5 ± 90.7 ^b^	61.1	4.5:1	3483.3 ± 2104.1 ^b^	81.2%

The data are presented as the mean ± standard deviation (SD). In each column, there is a significant difference between the data marked with different letters (*p* < 0.05), and there is no significant difference between the data marked with the same letter (*p* > 0.05), All values were estimated by ANOVA (Duncan’s post hoc) at *p* ≤ 0.05.

**Table 3 ijms-23-10942-t003:** The qRT-PCR primers of EmSAGs and internal reference genes *Em*-β-tubulin, *Em*-GAPDH.

Name	Primer	Primer Sequence (5′→3′)
β-tubulin	Forward	TCACTTTCGTCGGCAACTCAACC
	Reverse	AACTCCATCTCGTCCATACCCTCTC
GAPDH	Forward	GTGTAGCGGGCGTTTGAGGATG
	Reverse	GTCGTTCACAGCCACCACTTCC
SAG1	Forward	CGTGAAAGACTGGCAGAGAGGATTC
	Reverse	GGCGTCGTCGTAAGGATTGTAGAG
SAG2	Forward	CAGTGTAGTGACGCCGTGGAAG
	Reverse	ACTTGCTTGGGGTTCGCTTGTG
SAG10	Forward	CGCAACGGTTTATACATACGGTTTGG
	Reverse	CGGCATAGGCTGAGTTCTTGATGG
SAG11	Forward	CTCTACAATCCCGTCAGCGAAGC
	Reverse	TTTGTGGTATTGCGTGAGGGTAGC

**Table 4 ijms-23-10942-t004:** Primers of EmSAGs.

Name	Primer	Primer Sequence (5′→3′)
SAG10	Forward	ATGTGGATGTTCGCAGAA
	Reverse	TTAAAACAGAGCAAGTCCT
SAG11	Forward	ATGAGCGACCCTGTAACA
	Reverse	CTAGAATAGAAACGCAGCC

**Table 5 ijms-23-10942-t005:** Experimental grouping.

Group	Number of Rabbits	Inoculation Treatment (1 mL in Total)	Insect Attack (×10^5^)
Unchallenged control	8	Sterile PBS	--
Challenged control	8	Sterile PBS	1 × 10^5^ sporulated oocysts/week 4/oral
Quil-A saponin	8	Saponin (2 mg/mL)	1 × 10^5^ sporulated oocysts/week 4/oral
Recombinant pET-32a tag protein	8	recombinant pET-32a tag protein (150 μg) + saponin (2 mg/mL)	1 × 10^5^ sporulated oocysts/week 4/oral
rEmSAG10 immunized	8	SAG10 protein (150 μg) + saponin (2 mg/mL)	1 × 10^5^ sporulated oocysts/week 4/oral
rEmSAG11 immunized	8	SAG11 protein (150 μg) + saponin (2 mg/mL)	1 × 10^5^ sporulated oocysts/week 4/oral

## Data Availability

The datasets supporting the conclusions of this study are included in this article.

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

^ΔΔCT^ method. Methods.

