# Peer review of "Prokaryotic Expression of Eimeria magna SAG10 and SAG11 Genes and the Preliminary Evaluation of the Effect of the Recombinant Protein on Immune Protection in Rabbits"

_ijms, 2022, doi:10.3390/ijms231810942_

Round 1

Reviewer 1 Report

This is a report regarding the vaccine potential of of two SAG proteins from E. magna, SAG10 and SAG11, in rabbits. While the selection criteria, cloning and recombinant protein production are described clearly, the evaluation criteria regarding their usefulness as vaccine candidates unfortunately are unclear, thus significantly reducing the impact of the manuscript. Many of the conclusions described in the discussion section, particularly concerning immunology, are not valid. The manuscript is well written in good English in  general, but needs to be carefully checked for spelling errors and even more carefully in terms of technical terminology and phrasing. 

There are major issues with the methodology and discussion sections:

L 279: Transcription level of SAG genes
Please clarify this entire paragraph, it is impossible to determine how you have calculated the results shown in Fig 1 based on the paragraph as it stands now. If you used the DeltaDelta Ct method, please report how you calibrated your results. Also, it is very important that you check your primer sequences again, as the sequences do not match the genes nor do they match the organism you have worked with. This is a very significant concern.

L 155-156: Where is the data? This is not data shown in Table 1! 

L 164: Figure 5

Please clarify the legend: PET31a vector is the same as Trx-His-S tag? If that is so, is it correct to assume that you expressed and purified the Trx protein? Please explain your data in Figure 5 then. It is of concern, as it is inducing nearly the same level of specific IgG antibody as rEmSAG11 and that is a problem. 

Discussion:

L 219-220: Not a valid conclusion regarding IL-2 levels. 

L 223-225: Not a valid conclusion regarding IL-4 levels.

L 230-233: Not a valid conclusion based on your data. '

L 237-240: Not an accurate description of your results nor a valid conclusion. 

Author Response

Replies to Reviewer 1:

  • This is a report regarding the vaccine potential of of two SAG proteins from E. magna, SAG10 and SAG11, in rabbits. While the selection criteria, cloning and recombinant protein production are described clearly, the evaluation criteria regarding their usefulness as vaccine candidates unfortunately are unclear, thus significantly reducing the impact of the manuscript. Many of the conclusions described in the discussion section, particularly concerning immunology, are not valid. The manuscript is well written in good English in general, but needs to be carefully checked for spelling errors and even more carefully in terms of technical terminology and phrasing. 

Response: Thank you for your suggestion. We refer to Sufang Fang (evaluated by clinical symptoms, oocyst output and average weight gain) ((Fang, S., Gu, X., El-Ashram, S., Li, X., Yu, X., Guo, B.& Suo, X. Immune protection provided by a precocious line trivalent vaccine against rabbit Eimeria. Veterinary parasitology.2019, 275, 108927.) and Mohamed Sadek Bachene (evaluated by oocyst output and average weight gain) (Bachene, M. S., Temim, S., Ainbaziz, H., Bachene, A. & Suo, X.  A vaccination trial with a precocious line of Eimeria magna in Algerian local rabbits Oryctolagus cuniculus. Veterinary parasitology. 2018, 261 :73-76.) Evaluation index of Eimeria large precocious strain vaccine, in this basis, we increased the reduction rate of oocysts, the ratio of feed to meat and intestinal lesions. Moreover, we showed through animal experiments that, compared with the control group, the relative weight gain rates of rEmSAG10 and rEmSAG11 were 62.7% and 61.1%, respectively, their feed-to-meat ratios were 3.8:1 and 4.5:1, and their oocyst decrease rate were 70.8 %, 81.2%, and the intestinal lesions in the experimental group were significantly reduced. The immunology section in the discussion section has been revised (see L258-279), Spelling errors and terminology have been revised in the article.

  • There are major issues with the methodology and discussion sections:

L 279: Transcription level of SAG genes
Please clarify this entire paragraph, it is impossible to determine how you have calculated the results shown in Fig 1 based on the paragraph as it stands now. If you used the DeltaDelta Ct method, please report how you calibrated your results. Also, it is very important that you check your primer sequences again, as the sequences do not match the genes nor do they match the organism you have worked with. This is a very significant concern.

Response: Thank you for your suggestion. We screened two stable reference genes, Em-GAPDH and Em-β-tubulin. Finally, we used Em-β-tubulin as the reference gene for comparison. The qRT-PCR data were based on Kenneth Calculated by the method of J. Livak and Thomas D. Schmittgen (2-▲▲Ct), in the article Materials and methods we have supplemented how to proofread the results. We put the designed primer sequences (SAG1, SAG2, SAG10, SAG11) on NCBI through Primer BLAST https://www.ncbi.nlm.nih.gov/tools/primer-blast/index.cgi?LINK_LOC=BlastHome, the BLAST result is that the primer sequence is consistent with the gene. We used DNAMAN to align the primer sequences, all primer sequences matched the genes, we sent the full sequences of the genes EmSAG1, EmSAG2, EmSAG10 and EmSAG11 to Shanghai Sangon Bioengineering Co., Ltd. The sequences he designed were in our full sequence Part of the sequence, the amplified product sequence is about 80-300bp, and the sub-segment DNA represents our gene, and the qRT-PCR primer sequence is designed accordingly.

  • L 155-156: Where is the data? This is not data shown in Table 1! 

Response: Thank you for your suggestion, due to my own negligence, I mistakenly wrote Table 5 as Table 1, and Table 5 has been added in the article.

  • L 164: Figure 5

Please clarify the legend: PET32a vector is the same as Trx-His-S tag? If that is so, is it correct to assume that you expressed and purified the Trx protein? Please explain your data in Figure 5 then. It is of concern, as it is inducing nearly the same level of specific IgG antibody as rEmSAG11 and that is a problem. 

Response: Thank you for your suggestion, the pET32a plasmid is a prokaryotic expression vector, the C-terminal contains a 6×His tag, and the N-terminal contains a thrombin site, 6×His tag, TrXA site, Ek site and S tag, so The pET32a vector contains the Trx-His-S tag. Because our pET32a vector has a Trx-His-S tag and rEmSAG11 has a Trx-His-S tag, it is normal that it induces the same trend of specific IgG antibody levels as rEmSAG11, but rEmSAG11 has high IgG antibody levels in the Trx-His-S group.

  • Discussion:

L 219-220: Not a valid conclusion regarding IL-2 levels. 

L 223-225: Not a valid conclusion regarding IL-4 levels.

L 230-233: Not a valid conclusion based on your data. '

L 237-240: Not an accurate description of your results nor a valid conclusion. 

Response: Thanks for the suggestion, L 219-220, the discussion section on IL-2 levels has changed (see L258-263),

L 223-225, on IL-4 levels have been changed(see L268-276),

L 230-233, the conclusion section has been amended (see L441-443),

L 237-240, this part has been modified (see L277-279).

Reviewer 2 Report

Immunization against the rabbit coccidial pathogen Eimeria magna has received less attention than immunization against avian Eimeria lime E. tenella. In the current study, Pu et al. identified two potential SAGs that are highly expressed in merozoites. The two E. magna SAG proteins expressed in E.coli and as Trs-His-S tag fusion proteins were tested for immunoprotective effects. The methods mentioned in this manuscript are standard for studies of this kind. Similar but more intensive studies have been carried out in related species of Emeria, like E. tenella, so the subject is not very novel.

The manuscript needs to be improved in terms of background literature, detailed methods, and better presentation of results (figures and statistical analysis). The lack of significant IFN-γ /TGF-β response in the groups immunized with the recombinant SAG10/SAG11 proteins also needs to be explained.

A major revision addressing the key concerns is recommended for the manuscript.

Major Comments Listed below

Comment 1: Lack of significant IFN-γ /TGF-β response to recombinant SAG10/SAG11 immunization: There are many reports that Eimeria surface antigens (SAGs) induce inflammatory response involving IFN-γ / release. Liu et al, 2018 (Parasites Vectors 11, 325 (2018)) have reported significant IFN-γ / TGF-β production after immunization with recombinant E maxima SAG protein. In the data shown in Figure 6, no significant IFN-γ production was observed in the rEm-SAG10 and rEm-SAG11 injected animal groups. Although the authors claim there was a significant increase in TGF-B production in the results, the Figure 6 graph does not seem to substantiate this (Authors need to provide appropriate statistical pairwise comparisons and significance figures for this). It would be great if the authors could explain the lack of significant IFN-γ/ TGF-β response to recombinant SAG10/SAG11 injection.

Comment 2 - Figure 2- The purity of the recombinant protein ( lane 3) especially rEm-SAG11 in Figure 2b appears to be poor from the SDS PAGE, with too many other bands. Would these other proteins affect the immunogenicity/specificity?

Comment 3 - Figure 3 – The immunoblot image is not comprehensive and convincing: In the present image, each of the four lanes is cropped from four different western blots. Also, as the authors have used a large Trx – His- S-Tag protein fusion partner for producing the recombinant protein, there should be another control lane with the Trx – His- S-Tag protein alone as a control in another lane. Ideally, the western blot should feature three proteins rEm-SAG10, rEm-SAG11, and control (Trx – His- S-Tag protein) plus the protein marker in a single gel. And there should be two such blots displayed side by side – one probed with the Eimeria positive rabbit serum and the second with the Eimeria negative rabbit serum. Then only the specificity of the two recombinant proteins could be reliably verified.

Comment 5 - Figure 5 : It would be great if the authors can provide the statistical analysis and significance values for the data presented in Figure 5. The significance values between rEm-SAG11 and Trx – His- S-Tag protein in Figure 5(b) should be mentioned given the high variation in the Trx – His- S-Tag control protein sample OD readings

Comment 6 - Figure 6: Detailed statistical analysis for the cytokines in the serum is required for Figure 6. Especially pairwise comparison and significance values of rEm-SAG10 and rEm-SAG11 with other control samples.

Comment 7 – All figure legends: The authors are requested to improve the Figure legends. The legends should clearly explain the data presented without leaving too much to the reader's imagination. Appropriate statistical analysis detail is missing in many figure legends.

Other comments

Multiple Alignment of SAG protein sequence: If the nice if the authors could provide a figure featuring the multiple alignment of the Eimeria magna SAG1, SAG2, SAG10 and SAG11 protein sequences with orthologs from other well-studied Eimeria like E tenella, with salient features and predicted immunogenic epitopes marked.

Table 4 – Authors are requested to explain what they meant by Insect attack (x105) in the Table 4 header. Also, the Table heading ‘Quantity (piece), I feel refers to the number of animals in each group and needs to be appropriately modified.

Also, there are a few typos and other usage errors in the manuscript, which need to be addressed by the authors.

Author Response

Replies to Reviewer 2:

  • Immunization against the rabbit coccidial pathogen Eimeria magna has received less attention than immunization against avian Eimeria lime E. tenella. In the current study, Pu et al. identified two potential SAGs that are highly expressed in merozoites. The two E. magna SAG proteins expressed in E.coli and as Trs-His-S tag fusion proteins were tested for immunoprotective effects. The methods mentioned in this manuscript are standard for studies of this kind. Similar but more intensive studies have been carried out in related species of Emeria, like E. tenella, so the subject is not very novel.

Response: Thank you for your suggestion. Eimeria magna is a common coccidia in the rabbit intestine, which can cause anorexia, weight loss and diarrhea in rabbits. Severe infection will cause bloody stools, which will seriously affect the development of the rabbit industry. But drug resistance and drug residues in food are a growing problem. So rabbit coccidiosis vaccine is an effective means of prevention.

  • The manuscript needs to be improved in terms of background literature, detailed methods, and better presentation of results (figures and statistical analysis). The lack of significant IFN-γ /TGF-β response in the groups immunized with the recombinant SAG10/SAG11 proteins also needs to be explained.

Response: Thanks for the suggestion, the background literature (see L550-560 ), detailed methods (see L346-351 ) and results (see L113-125, L197, L216-219, L227-231) have been revised. The lack of a significant IFN-γresponse in the group immunized with recombinant SAG10/SAG11 protein has been explained in the article (see L264-267).

A major revision addressing the key concerns is recommended for the manuscript.

  • Comment 1: Lack of significant IFN-γ /TGF-β response to recombinant SAG10/SAG11 immunization: There are many reports that Eimeria surface antigens (SAGs) induce inflammatory response involving IFN-γ / release. Liu et al, 2018 (Parasites Vectors 11, 325 (2018)) have reported significant IFN-γ / TGF-β production after immunization with recombinant E maxima SAG protein. In the data shown in Figure 6, no significant IFN-γ production was observed in the rEm-SAG10 and rEm-SAG11 injected animal groups. Although the authors claim there was a significant increase in TGF-B production in the results, the Figure 6 graph does not seem to substantiate this (Authors need to provide appropriate statistical pairwise comparisons and significance figures for this). It would be great if the authors could explain the lack of significant IFN-γ/ TGF-β response to recombinant SAG10/SAG11 injection.

Response: Thanks for your suggestion, the article has supplemented with IFN-γ for no significant reason (see L264-267). TGF-β in revised Figure 6 was analyzed for significance and the result was a significant increase in the production of TGF-β (see appendix).

  • Comment 2 - Figure 2- The purity of the recombinant protein (lane 3) especially rEm-SAG11 in Figure 2b appears to be poor from the SDS PAGE, with too many other bands. Would these other proteins affect the immunogenicity/specificity?

Response: Thank you for your suggestion. The rEm-SAG11 in revised Figure 3b is a picture of the primary protein purification. In the end, our animal experiments used a single-band rEm-SAG11. The rEm-SAG11 in revised Figure 3b in the article has been changed.

  • Comment 3 - Figure 3 – The immunoblot image is not comprehensive and convincing: In the present image, each of the four lanes is cropped from four different western blots. Also, as the authors have used a large Trx – His- S-Tag protein fusion partner for producing the recombinant protein, there should be another control lane with the Trx – His- S-Tag protein alone as a control in another lane. Ideally, the western blot should feature three proteins rEm-SAG10, rEm-SAG11, and control (Trx – His- S-Tag protein) plus the protein marker in a single gel. And there should be two such blots displayed side by side – one probed with the Eimeria positive rabbit serum and the second with the Eimeria negative rabbit serum. Then only the specificity of the two recombinant proteins could be reliably verified.

Response: Thanks for your suggestion, we have supplemented the western blot plot of the control Trx-His-S-Tag protein. We supplements the western blot of the control Trx-His-S-Tag protein to prove that the two proteins rEm-SAG10 and rEm-SAG11 are immunogenic(see revised figure 4).

  • Comment 5 - Figure 5 : It would be great if the authors can provide the statistical analysis and significance values for the data presented in Figure 5. The significance values between rEm-SAG11 and Trx – His- S-Tag protein in Figure 5(b) should be mentioned given the high variation in the Trx – His- S-Tag control protein sample OD readings.

Response: Thank you for your suggestion, we have provided the data in revised Figure 6, and provided statistical analysis and significance values (see attachment), and also mentioned the significance value between rEm-SAG11 and Trx in the article (p <0.05).

  • Comment 6 - Figure 6: Detailed statistical analysis for the cytokines in the serum is required for Figure 6. Especially pairwise comparison and significance values of rEm-SAG10 and rEm-SAG11 with other control samples.

Response: Thank you for your suggestion, we have carried out a detailed statistical analysis of cytokines in serum in the attachment.

  • Comment 7 – All figure legends: The authors are requested to improve the Figure legends. The legends should clearly explain the data presented without leaving too much to the reader's imagination. Appropriate statistical analysis detail is missing in many figure legends.

Other comments.

Response: Thanks for your suggestion, we have supplemented the legend in the figure in detail.

  • Multiple Alignment of SAG protein sequence: If the nice if the authors could provide a figure featuring the multiple alignment of the Eimeria magna SAG1, SAG2, SAG10 and SAG11 protein sequences with orthologs from other well-studied Eimeria like E tenella, with salient features and predicted immunogenic epitopes marked.Table 4 – Authors are requested to explain what they meant by Insect attack (x105) in the Table 4 header. Also, the Table heading ‘Quantity (piece), I feel refers to the number of animals in each group and needs to be appropriately modified.Also, there are a few typos and other usage errors in the manuscript, which need to be addressed by the authors.

Response: Thank you for your suggestion, the multiple alignment of SAG protein sequences has been supplemented in the article (see figure 1), and the meaning of (x105) in the article has been revised. 

Round 2

Reviewer 1 Report

Please make an effort to correct the following:

Transcription level of SAG genes: L383-L401--The normalization of your data is not the issue I have. I am OK with you using the reference genes you have provided the data for, however, you need to provide data for the calibrator. How did you calibrate the result? What did you compare the expression levels with to obtain data that shows increased expression at certain developmental stages? This is not explained. The method you used to calculate your results is based on comparisons between exp. groups and control groups (that is the delta delta Ct method). I however do not think that you have a control group in this case--at least I am unable to identify the control group. Please explain. The reference you provided is fine, I am familiar with it, and it does state that you need a comparison group to perform delta delta Ct.

Also, the primer sequences match ONLY the ON468435.1 sequence, which is the one you provided to GenBank yourself. There is no match to any other Eimeria sequence in Genbank. Please explain.

Please provide an explanation for your reply:

"Because our pET32a vector has a Trx-His-S tag and rEmSAG11 has a Trx-His-S tag, it is normal that it induces the same trend of specific IgG antibody levels as rEmSAG11, but rEmSAG11 has high IgG antibody levels in the Trx-His-S group."

If the Trx-His-S tag induces the same level of SPECIFIC IgG ANTIBODY as rEmSAG11 with a Trx-His-S tag, please explain what the antibody is specific to. This is exactly my concern. If you get rid of the tag, based on the data you provide, I believe that the levels of specific IgG antibody that will be induced by the rEmSAG11 or in fact rEmSAG10 alone are VERY LOW. 

Author Response

1)Please make an effort to correct the following:

Transcription level of SAG genes: L383-L401--The normalization of your data is not the issue I have. I am OK with you using the reference genes you have provided the data for, however, you need to provide data for the calibrator. How did you calibrate the result? What did you compare the expression levels with to obtain data that shows increased expression at certain developmental stages? This is not explained. The method you used to calculate your results is based on comparisons between exp. groups and control groups (that is the delta delta Ct method). I however do not think that you have a control group in this case--at least I am unable to identify the control group. Please explain. The reference you provided is fine, I am familiar with it, and it does state that you need a comparison group to perform delta delta Ct.

Also, the primer sequences match ONLY the ON468435.1 sequence, which is the one you provided to GenBank yourself. There is no match to any other Eimeria sequence in Genbank. Please explain.

Respond: Thank you for your constructive comments. The expression levels in this study were compared based on the unsporulated stage, which served as the control ground. This information has been added in Methods section (see L356-359). For the second question, you pointed out that our primer sequences only match the sequence of EmSAG1 (accession number ON468435.1) which we submitted to GenBank. For one thing, this  indicated that our primer sequences are specific, and this primer sequence can only amplify the EmSAG1 sequence. For another, the similarity of the sequences of SAGs genes ranges from 26.26% to 38.76%, they were defined as SAGs genes as they all have conserved GPI anchor points. Although there are no hits for primer blast except the EmSAG1 we submitted, while we used the whole sequences as the query to do nucleotide blast, there were more than 50s SAGs sequences of other Eimeria sequence in Genbank were shown, including Eimeria tenella (26.26% identity to EmSAG1), Eimeria mitis (35.32% identity to EmSAG1),  Eimeria brunetti (33.17% identity to EmSAG1) and so on. Therefore, as the highest sequence similarity is no more than 40%, the primer sequences match ONLY the ON468435.1 sequence we provided is make sense.

2)Please provide an explanation for your reply:

"Because our pET32a vector has a Trx-His-S tag and rEmSAG11 has a Trx-His-S tag, it is normal that it induces the same trend of specific IgG antibody levels as rEmSAG11, but rEmSAG11 has high IgG antibody levels in the Trx-His-S group."

If the Trx-His-S tag induces the same level of SPECIFIC IgG ANTIBODY as rEmSAG11 with a Trx-His-S tag, please explain what the antibody is specific to. This is exactly my concern. If you get rid of the tag, based on the data you provide, I believe that the levels of specific IgG antibody that will be induced by the rEmSAG11 or in fact rEmSAG10 alone are VERY LOW.

Respond: Thank you for your constructive comments. We are sorry that we haven’t explained this question clearly. The statement we made in "Because our pET32a vector has a Trx-His-S tag and rEmSAG11 has a Trx-His-S tag, it is normal that it induces the same trend of specific IgG antibody levels as rEmSAG11, but rEmSAG11 has high IgG antibody levels in the Trx-His-S group." is not right. As our recombinant proteins were ligated to pET32a(+) Vector and prokaryotic expressed in BL21(DE3) cells, they included an approximately 20 kDa epitope tag fusion peptide, while the molecular weight of these two SAG proteins were predicted to be 21 kDa and 22 kDa, therefore the two recombinant proteins were at ~41 kDa and ~42 kDa. Because the ~20 kDa epitope tag fusion peptide accounts for 1/2 of the molecular weight of the recombinants, it is necessary for us to include this ~20 kDa epitope tag fusion peptide as a control group, which we used "pET32a " to represent it (seeL130). As both “pET32a ” and the two recombinant proteins are foreign proteins for our host (rabbits), they were likely to induce a certain level of antibody specific to the protein they vaccinated. Because we detected the specific antibody of the recombinant protein by using the recombinant protein as antigen, but the recombinant protein included the "~20 kDa epitope tag fusion peptide", which caused the certain levels of IgG antibody as you can see in Figure 6. We totally agree with your concern that "If you get rid of the tag, based on the data you provide, I believe that the levels of specific IgG antibody that will be induced by the rEmSAG11 or in fact rEmSAG10 alone are VERY LOW. ", however, the difference between immunization group were significant; according to significant different effects between the "pET32a " group and immunization group in other criteria that could reflect the effects of vaccines, including the body weight gain, oocyst reduction ratio, feed conversion ratio. We believe that the effect of our pure SAG10 and SAG11 proteins can protect the rabbits from the infection of E. magna. A similar strategy was also recently used in the chicken coccidia (Krishnedu Kundu, Vet Parasitol, 2017).

Reviewer 2 Report

The authors have edited the manuscript by addressing most of the comments in the first round. The statistical analysis of the change in TGF-β expression was carried out and was found to be significant. The authors have also discussed the lack of significant change in IFN-γ response in the challenge experiments. Statistical analysis of the change in the specific rEm-SAG10/SAG11 antibodies and cytokines in the serum has also been incorporated in the revised version. (Also a new multiple alignment figure with various Eimeria SAG proteins has also been provided.

For figure 4, (Western blot) It would have been more conclusive if the authors had provided a new figure with all protein in a single gel, rather than cropped lanes.

Some minor errors are still noted ( Kindly correct these before the final version) :

Two Figures with Figure 6 numbering -Please note that there are now two figures with numbering Figure 6, kindly correct the number of the last figure to Figure 6).

Line 144 - rabbit-negative serum did not recognize rEmSAG10 or rEmSAG11 (Fig. 4, lanes 3 and 4)

I feel it should be corrected to Fig. 4, lanes 4 and 5).

I recommend the manuscript for publication.

Author Response

1)The authors have edited the manuscript by addressing most of the comments in the first round. The statistical analysis of the change in TGF-β expression was carried out and was found to be significant. The authors have also discussed the lack of significant change in IFN-γ response in the challenge experiments. Statistical analysis of the change in the specific rEm-SAG10/SAG11 antibodies and cytokines in the serum has also been incorporated in the revised version. (Also a new multiple alignment figure with various Eimeria SAG proteins has also been provided.

For figure 4, (Western blot) It would have been more conclusive if the authors had provided a new figure with all protein in a single gel, rather than cropped lanes.

Respond:Thank you for your positive comments for our revised manuscript. For the issue you mentioned for Figure 4, because we performed the Western blot of the two recombinant protein separately, including recombinant protein probed with positive serum and negative serum (they must incubate the recombinant protein separately, and generate two blots), to make the Figure more clear to readers, we cropped lanes and combine them in one Figure. For each time we performed Western blot, we had protein marker along with these two recombinant proteins, therefore we cropped lanes according to the markers; we have uploaded the original blots to the submission system for review.

2)Some minor errors are still noted ( Kindly correct these before the final version) :

Two Figures with Figure 6 numbering -Please note that there are now two figures with numbering Figure 6, kindly correct the number of the last figure to Figure 6).Line 144 - rabbit-negative serum did not recognize rEmSAG10 or rEmSAG11 (Fig. 4, lanes 3 and 4)I feel it should be corrected to Fig. 4, lanes 4 and 5).I recommend the manuscript for publication.

Rspond:Thank you for careful reading of our manuscript and pointing the erros out. For the two Figures with Figure 6 numbering, we have corrected the number of  the last Figure 6 to Figure 7.For the typing error in Line 144 - Rabbit negative serum does not recognize rEmSAG10 or rEmSAG11 (Figure 4, lanes 3 and 4),it has been  changed to “(Figure 4, lanes 4 and 5)”.

Round 3

Reviewer 1 Report

Thank you for your efforts. The changes you made have improved the manuscript slightly.